# Engineering of Silica Mesoporous Materials for CO_2_ Adsorption

**DOI:** 10.3390/ma16114179

**Published:** 2023-06-04

**Authors:** Oyundari Tumurbaatar, Margarita Popova, Violeta Mitova, Pavletta Shestakova, Neli Koseva

**Affiliations:** 1Institute of Organic Chemistry with Centre of Phytochemistry, Bulgarian Academy of Sciences, Acad. G. Bonchev St., Bl. 9, 1113 Sofia, Bulgaria; oyundari.tumurbaatar@orgchm.bas.bg (O.T.); margarita.popova@orgchm.bas.bg (M.P.); pavletta.shestakova@orgchm.bas.bg (P.S.); 2Institute of Polymers, Bulgarian Academy of Sciences, Acad. G. Bonchev St., Bl. 103A, 1113 Sofia, Bulgaria; mitova@polymer.bas.bg; 3Bulgarian Academy of Sciences, №1, 15 Noemvri St., 1040 Sofia, Bulgaria

**Keywords:** CO_2_ capture, modified mesoporous silicas, DOPO derivative

## Abstract

Adsorption methods for CO_2_ capture are characterized by high selectivity and low energy consumption. Therefore, the engineering of solid supports for efficient CO_2_ adsorption attracts research attention. Modification of mesoporous silica materials with tailor-made organic molecules can greatly improve silica’s performance in CO_2_ capture and separation. In that context, a new derivative of 9,10-dihydro-9-oxa-10-phosphaphenanthrene-10-oxide, possessing an electron-rich condensed aromatic structure and also known for its anti-oxidative properties, was synthesized and applied as a modifying agent of 2D SBA-15, 3D SBA-16, and KIT-6 silicates. The physicochemical properties of the initial and modified materials were studied using nitrogen physisorption and temperature-gravimetric analysis. The adsorption capacity of CO_2_ was measured in a dynamic CO_2_ adsorption regime. The three modified materials displayed a higher capacity for CO_2_ adsorption than the initial ones. Among the studied sorbents, the modified mesoporous SBA-15 silica showed the highest adsorption capacity for CO_2_ (3.9 mmol/g). In the presence of 1 vol.% water vapor, the adsorption capacities of the modified materials were enhanced. Total CO_2_ desorption from the modified materials was achieved at 80 °C. The obtained silica materials displayed stable performance in five CO_2_ adsorption/desorption cycles. The experimental data can be appropriately described by the Yoon–Nelson kinetic model.

## 1. Introduction

The increasing volumes of carbon dioxide released into the atmosphere by human activities (greenhouse effect) are considered one of the global concerns of society [1]. Carbon dioxide is among the main constituents of greenhouse gases, being both of natural and anthropogenic origin [2]. Generally, due to the presence of greenhouse gases in the atmosphere, the temperature of the Earth is conducive to living systems. However, over the last decades, a steady rise in the average temperature of the planet has been recorded, and the trend is expected to continue, reaching an increase of 6.4% by the end of this century [3]. The anthropogenic greenhouse effect contributes significantly to climate change and extreme phenomena such as floods and droughts [2]. Therefore, a reduction in the CO_2_ emissions into the atmosphere would mitigate the adverse effect of human activity and industrialization in particular [3]. On the other hand, CO_2_ is an abundant resource for many chemical industries, and its capture and recycling provide ample opportunities for research [4].

Various CO_2_ capture and separation technologies have been under development. Among them, the adsorption or absorption approaches applied separately or in conjunction with membrane separation and chemical-looping combustion have been considered as industrially relevant technologies [5]. The adsorption methods are characterized by excellent selectivity and low consumption of energy. Two main classes of adsorbents are recognized: physisorbents [6] (zeolite, carbons, Si/Al gel) and chemisorbents (amines [7], caustics, metal oxides [8,9]). Intensive investigations have been focused on materials applicable in CO_2_ physical adsorption, including carbon-based materials [10,11,12], zeolites [13,14], and mesoporous silicas [15,16,17]. The latter are attractive materials containing a large number of micro- and mesopores and plenty of reactive Si-OH groups on the pore surface, which afford adsorption sites and present suitable centers for functionalization. Thus, silica’s features for efficient CO_2_ capture can be greatly enhanced via surface modification of the material [18,19,20,21,22]. Different amines have been immobilized onto mesoporous silicas chemically [23] or via hydrogen bonds [24] and studied in CO_2_ capture and separation experiments [25,26,27]. The modified materials displayed combined physi- and chemisorption resulting in an improved adsorption capacity and selectivity [28,29,30]. Amino silane derivatives have evolved as appropriate agents for feasible post-grafting of SBA-15 material under mild conditions [31]. The same material modified with hyperbranched amino grafts adsorbed CO_2_ reversibly with very high CO_2_ capacity values above 3 mmol CO_2_/g at ambient temperature [32]. Similar results were found for amine-loaded KIT-6, displaying a correlation between the sorbent capacity and the amount of amine loaded [27]. SBA-16 material grafted with N-(2-aminoethyl)-3-aminopropyltrimethoxysilane was characterized by good hydrothermal stability and a moderate adsorption capacity of about 0.73 mmol/g adsorbent [33]. The impregnation of mesoporous silica with amines has been applied to obtain adsorbents that do not lose their capacity in the presence of moisture; on the contrary, moisture improves their capacity [34].

The published data about amine-functionalized zeolites, MOF, and carbons have also revealed a high CO_2_ adsorption capacity—between 40 and 80 mg/g at 1 atm and 25 °C [35]. Due to its high resistance to moisture and thermal stability, porous alumina has also been studied as a potential adsorbent for CO_2_ after modification with organic amines. Aluminum oxide impregnated with 40% tetraethylenepentamine showed a capacity of 1.83 mmol/g at 25 °C [36]. Modified with the same polyamine, commercial Y zeolite (Si/Al = 60) achieved an adsorption capacity of 4.27 mmol/g at 60 °C in a gas stream of 15% CO_2_ and 7% water vapor [37]. A total regeneration after 10 consecutive cycles of adsorption/desorption was obtained on beta zeolite impregnated with tetraethylenepentamine [38].

Besides amino modification, the introduction of other functionalities onto the sorbent surface could also enhance the CO_2_ adsorption capacity and/or selectivity [39]. In our recent study [40], novel (3-aminopropyl)triethoxysilane (APTES) derivatives with tailored structures, which included an azomethine group conjugated with a furanyl ring or an aminophosphonate fragment, were used to modify the pore surface of MCM-48 and SBA-15 silicas. All the modified samples displayed a 1.5- to 2-fold increase in the capacity for CO_2_ adsorption in comparison to the initial materials. The MCM-48 material grafted with Schiff base residues was more efficient in the CO_2_ capture experiments than the analogous SBA-15 sample or those bearing aminophosphonate fragments [40]. Microporous polycarbazole, which possessed an electron-rich scaffold, showed an excellent uptake capacity for carbon dioxide and good selectivity toward CO_2_ over N_2_ [41]. Enhanced sorption capacity toward CO_2_ (4.9 mmol·g^−1^) under 367 K and 1 bar partial pressure of CO_2_ was measured for a ceramic material coated with MgO/carbon nanofibers composite [42].

The present paper reports the results of the preparation of a new derivative of APTES and its use in the post-synthetic modification of KIT-6, SBA-15, and SBA-16 mesoporous silicas. 9,10-Dihydro-9-oxa-10-phosphaphenanthrene-10-oxide (DOPO) has been selected as the modifying molecule due to its reactivity and ability to improve the anti-oxidative and flame-retarding properties of materials [43]. Moreover, it possesses an electron-rich aromatic structure that favors interactions with the CO_2_ molecule. The performance of the modified silica adsorbents has been evaluated and compared to the initial materials in CO_2_ adsorption experiments carried out under dynamic and static conditions.

## 2. Materials and Methods

### 2.1. Materials

Pluronic P123 (≥98%) (Sigma-Aldrich Chemie, Schnelldorf, Germany), (3-aminopropyl) triethoxysilane (APTES) (≥98%) (Sigma-Aldrich, Saint Louis, MO, USA), 9,10-dihydro-9-oxa-10-phosphaphenanthrene-10-oxide (DOPO) (≥98%) (Sigma-Aldrich Chemie, Schnelldorf, Germany), hexadecyltrimethylammonium bromide (CTAB) (≥98%) (Sigma-Aldrich Chemie, Schnelldorf, Germany), tetraethyl orthosilicate (TEOS) 98% (Fisher Scientific, Loughborough, UK), hydrochloric acid (37%) (Valerus, Bulgaria), and sodium hydroxide (NaOH) (Merck, Darmstadt, Germany) were used without further purification. Furfural (≥98%) (Sigma-Aldrich Production GmbH, Product Brand Fluka, Buchs, Switzerland) was freshly distilled before use. The solvents used (toluene and 2-methoxyethyl ether (diglyme) (Sigma-Aldrich, Saint Louis, MO, USA)) were dried via standard procedures. Deionized distilled water was used in the preparation of all solutions.

### 2.2. Preparation of KIT-6, SBA-15, and SBA-16

The mesoporous KIT-6, SBA-15, and SBA-16 silicas were prepared through hydrothermal synthesis. The synthesis of KIT-6 [44] includes the following steps: (i) Pluronic 123 (12 g, 2.1 mmol) was added at room temperature to an aqueous medium prepared from 37% HCl (37.1 mL) in 366 mL H_2_O. (ii) The nano-colloidal solution was mixed with butanol (15 mL), and 24 g of tetraethyl orthosilicate (TEOS) was added to the clear solution in 1 h. After that, the mixture was aged at 140 °C for 24 h under static conditions. (iii) The obtained solid product was filtered without further washing and dried under vacuum at 100 °C overnight. (iv) The material was calcined in air at 550 °C for 6 h with a heating rate of 1 K/min.

The synthesis of SBA-15 [45] was performed using the hydrothermal procedure. Pluronic P123 (6 g, 1.0 mmol) was dissolved in 180 g distilled water and 18.5 g 37% HCl at room temperature. Next, 12 g TEOS was added to the solution and stirred at 35 °C for 24 h. The obtained gel was transferred to a Teflon vessel jacketed in a stainless-steel autoclave and heated in an oven at 95 °C for 24 h. The obtained white material was filtered and calcined in an oven in the air at 290 °C for 2 h and at 550 °C for 5 h with a heating rate of 1 K/min. 

SBA-16 was prepared according to the procedure of Hu et al. [46]. Pluronic F127 triblock copolymer and CTAB were used as templates and tetraethylorthosilicate (TEOS) as the silica source. F127 (4 g, 0.3 mmol) and CTAB (0.48 g, 1.3 mmol) were dissolved entirely into a solution of 130 mL water and 10 mL concentrated HCl. Under continuous stirring, 4.0 mL TEOS was added. The mixture was stirred at 40 °C for 1 h and then the temperature was increased to 80 °C for 24 h under static conditions. The obtained material was filtrated, washed with water three times, and dried in an oven at 50 °C. The white material was calcined in air at 550 °C for 5 h with a heating rate of 1 K/min.

### 2.3. Preparation of DOPO Derivative of APTES

The preparation procedure included firstly the synthesis of a Schiff base (denoted as SAPTES) from the reaction of APTES and furfural as described in our recent publication [40]. The procedure and the NMR and IR data are provided in the Appendix A.

Secondly, the reaction of SAPTES (1.00 g, 3.1 mmol) with DOPO (0.68 g, 3.1 mmol) dissolved in diglyme (5 mL) was carried out in the presence of a catalyst CdI_2_ (0.023 g, 0.06 mmol). The reaction proceeded for 24 h at 50 °C under constant stirring. Then, the mixture was vacuumed at 45 °C on a rotavapor for 10 h to remove the solvent. The product was assigned as DAPTES. It contained about 22 wt% diglyme.

^1^H NMR (599.98 MHz, DMSO) δ ppm: 8.19–7.10 (m, 8H from DOPO fragment and 1H from furanyl ring); 6.45–6.31 (m, 2H, from furanyl ring); 4.50–4.23 (m, 1H, -HNCHP-); 3.67–3.42 (m, 6H, CH_3_CH_2_O–); 2.36–2.23 (m, 2H, –CH_2_CH_2_CH_2_NH–); 1.28–1.22 (m, 2H, –CH_2_CH_2_CH_2_NH–); 1.11–1.06 (m, 9H, CH_3_CH_2_O–); 0.34–0.31 (m, 2H, –CH_2_Si–). The signals at 3.50 ppm, 3.42 ppm, and 3.24 ppm belong to the residual solvent (diglyme).

^13^C NMR (150.87 MHz, DMSO) δ ppm: 149.73, 143.44, 132.23 (s), 131.08 (d), 128.62 (d), 125.96 (d), 124.95 (d), 124.38 (d), 120.30 (d), 111.10 (d), 109.92 (d), 109.60 (d)—signals from DOPO fragment and furanyl ring; 58.03 (s, CH_3_CH_2_OSi–); 55.95 (d, ^1^J = 115.17 Hz, –HNCHP(O)–); 50.74 (s, –CH_2_CH_2_CH_2_NH–); 22.84 (s, -CH_2_CH_2_CH_2_NH-); 18.73 (s, CH_3_CH_2_OSi–); 7.49 -CH_2_Si–).

^31^P{H} NMR (242.88 MHz, DMSO) δ ppm: 30.21, 28.19, and 27.99.

IR (cm^−1^): 3000–2800 (ν (C–H)); 1450 (ν (C–C) in aromatic rings); 1242 (ν (P=O)); 1150-1050 (ν (C–O) and ν (Si–OCH_2_)); 758 (γ(FurC–H and ν (Si–CH_2_)).

### 2.4. Modification of Mesoporous Silicas with DAPTES

SBA-15, SBA-16, or KIT-6 (1.0 g each of samples) was heated in an oven at 120 °C for 1 h. The hot mesoporous silica was dispersed in toluene (50 mL). After that, DAPTES (0.155 g) was added to the dispersion, which was stirred for 24 h at 60 °C on a magnetic stirrer. The obtained material was washed three times with chloroform in order to remove the unreacted modifying agent or residual diglyme. The modified samples were denoted as SBA-15/DAPTES, SBA-16/DAPTES, and KIT-6/DAPTES.

### 2.5. Methods

#### 2.5.1. Material Characterization

NMR spectra were recorded on a Bruker AVANCE II+ 600 NMR spectrometer (Bruker, Germany) operating at 599.98 MHz ^1^H frequency (150.88 for ^13^C, 242.88 for ^31^P, and 119.19 MHz for ^29^Si). The instrument is equipped with a dual direct broadband 1H/109Ag-^31^P probe head with a z-gradient. Deuterated chloroform or DMSO was used as the solvent, and the chemical shifts were calibrated to a residual solvent peak. In the solid-state measurements, a 4 mm solid-state CP/MAS dual ^1^H/X probe head was used. The samples were loaded in 4 mm zirconia rotors and spun at a magic angle spinning (MAS) rate of 10 kHz for all measurements. The quantitative ^29^Si NMR spectra were recorded with a single-pulse sequence, 90° pulse length of 4.5 μs, 3K time domain data points, spectrum width of 29 kHz, 1024 scans, and a relaxation delay of 60 s. The spectra were processed with an exponential window function (line broadening factor 100) and zero-filled to 16 K data points. The ^1^H→X cross-polarization MAS (CP MAS) spectra were acquired with the following experimental parameters: ^1^H excitation pulse of 3.6 μs, 5 ms contact time for the ^1^H→^29^Si and 2 ms for the ^1^H→^13^C CP experiments, and 5 s relaxation delay; typically, 6000 scans were accumulated for the ^1^H→^29^Si CPMAS experiments and up to 5000 scans for the ^1^H→^13^C CP MAS spectra. The ^1^H SPINAL-64 decoupling scheme was used during the acquisition of CP experiments.

ATR–FTIR spectra were recorded using an IRAffinity-1 “Shimadzu” Fourier-transform infrared (FTIR) spectrophotometer (Shimadzu, Kyoto, Japan) with a MIRacle attenuated total reflectance attachment. For each spectrum, a resolution of 4 cm^−1^ and 50 scans were applied. IRsolution software was used to process the collected spectral data.

The thermogravimetric measurements were obtained with a STA449F5 Jupiter of NETZSCH Gerätebau GmbH (Netzsch, Selb, Germany) in the temperature interval up to 600 °C, with a 5 °C/min heating rate in air flow, followed by a hold-up of 1 h.

The specific surface by Brunauer, Emmett, and Teller (BET), the diameter, and the pore size distribution of the obtained materials were determined by nitrogen adsorption at −196 °C in the pressure range p/p_0_ = 0.001–0.999, using an analyzer “AUTOSORB iQ-MP/AG” (Anton Paar GmbH, Graz, Austria). The samples were degassed at 80 °C with a heating rate of 5 °C/min for 16 h before every measurement.

#### 2.5.2. CO_2_ Adsorption

The CO_2_ adsorption experiments were carried out in dynamic conditions in a flow system. Prior to the start of the adsorption experiments, the adsorbent sample (0.40 g) was dried at 150 °C for 1 h. The experiments for CO_2_ adsorption were performed at 25 °C with 3 vol.% CO_2_/N_2_ at a flow rate of 30 mL/min. The samples were pressed and crushed in order to obtain materials with particle sizes of 0.2–0.8 mm. Additionally, the experiments for CO_2_ and water vapor adsorption (3 vol.% CO_2_ plus 1 vol.% water vapor) were performed at a 30 mL/min flow rate. The adsorption capacities of the materials were calculated based on the amounts of adsorbed CO_2_ and water vapor by using online GC analysis (gas chromatograph NEXIS GC-2030 ATF (Shimadzu, Kyoto, Japan) with a 25 m PLOT Q capillary column).

The CO_2_ adsorption measurements under static conditions were determined at 0 °C, 25 °C, and 50 °C with an AUTOSORB iQ-MP-AG (Anton Paar GmbH, Graz, Austria) surface area and pore size analyzer (from Quantachrome, Anton Paar GmbH, Graz, Austria). The Quantachrome software was used for the transformation of the primary adsorption data.

## 3. Results and Discussion

### 3.1. Synthesis and Characterization of DAPTES

The synthetic procedure to obtain a DOPO derivative of APTES could include different reaction routes. DOPO possesses a hydrogen phosphonate moiety that is reactive toward amines via the Atherton–Todd reaction [47,48]. However, this reaction proceeds in the presence of a base, i.e., tertiary amines, as scavengers of the liberated HCl. The side product, most often triethyl amine hydrochloride, should be removed from the reaction mixture, which is a laborious process that decreases the yield. Therefore, based on our previous experience, the synthetic path applied included the reaction of DOPO with a Schiff base (denoted as SAPTES) derived from APTES. The preparation of SAPTES has been reported in our previous paper [40]. SAPTES was obtained quantitatively from the reaction of APTES with furfural in toluene. The solvent was removed, and the Schiff base was subjected to a reaction with DOPO. The reagents were taken in an equimolar ratio. The interaction proceeded under mild conditions, i.e., stirring the reaction mixture at 50 °C for 24 h in the presence of cadmium iodide as catalyst (Figure 1). The formation of the product denoted as DAPTES was proven by NMR analysis including ^1^H, ^13^C{^1^H}, and ^31^P{^1^H}, as well as 2D experiments ^1^H-^1^H COSY, ^1^H-^13^C HSQC, and ^1^H-^13^C HMBC spectra (see Appendix A).

The ^1^H NMR spectrum of DAPTES (Appendix A) shows a complex spectral pattern, particularly in the aromatic region. We suggest that this is due to the structural complexity of DAPTES including asymmetric carbon atoms and bulky groups, which result in the possible coexistence of at least two isomers in the solution. The ^1^H NMR spectrum of DAPTES was compared to that of SAPTES. The assignment of the signals for the two compounds is provided in the Materials and Methods section. A multiplet between 4.50 ppm and 4.23 ppm is seen in the ^1^H NMR spectrum of DAPTES as proof of the formation of the -HNCH(Fur)P(O)- motif. The signal for the hydrogen from the imine -N=CH-group in SAPTES appears at 8.00 ppm, i.e., in the spectral region of the protons of DOPO. Therefore, it is not straightforward to conclude that this signal disappeared in the DAPTES spectrum due to its possible overlap with the DOPO resonances. However, it is clearly seen that the signal for the carbon atom from the imine group at about 152 ppm in the ^13^C NMR spectrum of SAPTES disappeared, and a doublet at 55.95 ppm with a coupling constant ^1^J{PC} = 115.17 Hz appeared in the ^13^C NMR spectrum of DAPTES (Appendix A) due to the newly formed aminophosphonate structure (–HNCH(Fur)P(O)–). Further evidence for the completion of the addition reaction is found in the comparison of the ^1^H and ^31^P{H} NMR spectra of DOPO and that of DAPTES. The doublet at 8.07 ppm (^1^J{PH} = 594 Hz) due to the P-H bond in the ^1^H NMR spectrum of DOPO (Appendix A) did not register in the ^1^H NMR spectrum of DAPTES as a consequence of the addition of the P-H bond from DOPO to the imine function of SAPTES. Secondly, the signal for the phosphorus atom in DOPO shifted downfield by 14 ppm in the ^31^P{H} NMR spectrum of DAPTES, as seen in Figure 1. Additionally, the DEPT-135 experiment (Appendix A) was performed, and the 2D ^1^H-^1^H COSY (Appendix A), ^1^H-^13^C HSQC (Appendix A), and ^1^H-^13^C HMBC (Appendix A) spectra of DAPTES were registered to further confirm that the product presents one substance (one and the same composition and atoms connectivity), i.e., the complexity of the spectra is due to conformational isomers.

The IR spectrum of DAPTES is shown in Figure 2. Characteristic bands of the oxa-10-phosphaphenanthrene-10-oxide residue are present: at 1450 cm^−1^ attributed to the aromatic rings and at 1242 cm^−1^ due to P=O stretching, and the absorption in the region 960–930 cm^−1^ can be assigned to the P-O-phenyl vibrations [48,49]. The lack of a band at about 2385 cm^−1^, characteristic of the P-H stretching, also confirmed the completion of the addition reaction of the P-H bond from DOPO to the imine group of SAPTES. The broad absorption in the region 3000–2800 cm^−1^ arises from C-H stretching in the alkyl and aromatic fragments of the DAPTES molecule. The intensive band in the spectral region 1150–1050 cm^−1^ is attributed to C-O-C stretching in the furanyl ring and Si-OCH_2_ stretching in the triethoxysilicate moiety [50]. A band due to C-H deformations of the furanyl ring overlapping with Si-C stretching is also seen at 758 cm^−1^ [51,52]. Both NMR and IR spectral analyses provided evidence for a complete conversion of the reaction of SAPTES with DOPO.

### 3.2. Modification of KIT-6, SBA-15, and SBA-16 Silica Materials

The SBA-15 or SBA-16 and KIT-6 materials were heated at 120 °C for 2 h to remove the adsorbed humidity and then dispersed in dry toluene. After that, DAPTES (15.5% of the silica weight) was added to each of the dispersions and allowed to react with the surface -SiOH groups for 24 h at 60 °C. The modified samples were denoted as SBA-15/DAPTES, SBA-16/DAPTES, and KIT-6/DAPTES.

The amounts of DAPTES grafted onto the KIT-6, SBA-15, and SBA-16 pore surface were calculated by using TGA analysis. The modified samples showed a weight loss between 12.5 and 13.7 wt% at temperatures above 250 °C up to 600 °C (Appendix A). The comparison of the initial and the modified sorbents revealed an additional weight lost between 6.3 and 9.4 wt% for the latter, which was attributed to the decomposition of the grafted moieties. These results indicate approximately similar grafting degrees of DAPTES in the modified mesoporous silicas. Moreover, the modification resulted in the hydrophobization of the pore surface of the sorbents as seen by the comparison of the TG curves in the temperature range 50–150 °C for the parent and grafted samples (Appendix A).

The IR spectra (Figure 2) of all the sorbent-modified samples show strong absorption in the spectral interval 1100–950 cm^−1^ attributed to Si-O-Si stretching vibrations of the silica matrix. Besides the dominating absorption of the latter, the presence of shoulders at about 1240 cm^−1^, 950–930 cm^−1^, and at 758 cm^−1^ evidenced the presence of DOPTES residues on the pore surface of the silica matrix.

The successful modification of the mesoporous silicas with DAPTES was also evidenced by ^1^H→^29^Si CP MAS (Figure 3) and ^1^H→^13^C cross polarization magic angle spinning (CP-MAS) spectra. The CP technique is based on the transfer of magnetization from abundant spins (^1^H) to low-sensitivity nuclei (^29^Si or ^13^C) via through-space dipole–dipole interactions. In the ^1^H→^29^Si CP MAS spectra, this method gives selective enhancement of the resonances from ^29^Si units with ^1^H in their vicinity, such as Si-OH groups or Si-sites with attached organic functional groups. In the ^1^H→^29^Si CP MAS spectra of the modified samples, a broad, low-intensity spectral pattern centered at around −57 ppm characteristic for T^2^ type ((SiO)_2_SiOH-R; R: DAPTES residue) organosiloxane moieties was detected, in addition to the Q^4^ (−110 ppm), Q^3^ (−100 ppm), and Q^2^ (−90 ppm) resonances that are typically observed in the spectra of non-modified silicas (Appendix A).

The ^1^H→^13^C CP MAS spectra (Appendix A) of SBA-16/DAPTES, KIT-6/DAPTES, and SBA-15/DAPTES show a typical spectral pattern in the region 155–110 ppm characteristic of the aromatic and furan moieties as well as the resonances for carbon atoms from the Si-**C**H_2_**C**H_2_**C**H_2_-NH-**C**H-P(O) structural motif (Si-CH_2_- at 9 ppm, -CH_2_- at 22 ppm, -CH_2_-NH-CH- from 50 to 60 ppm). Resonances from the residual -Si(OCH_2_CH_3_)_3_ structural fragments of DAPTES were also detected at around 62 ppm (-OCH_2_) and 18 ppm (-CH_3_), indicating that not all three OCH_2_CH_3_ groups have reacted with the silanol groups from the mesoporous silicas. As an example, Appendix A shows the ^1^H→^13^C CP MAS spectra of the KIT-6/DAPTES and SBA-15/DAPTES samples.

The textural properties of the initial and modified materials were determined using N_2_ adsorption/desorption measurements. The isotherms displayed type-IV isotherms according to the IUPAC classification, which is typical for mesoporous silicas [53]. They are presented in Figure 4, and the calculated parameters are listed in Table 1.

The calculated values for the specific surface area of the initial materials decreased in the following order: KIT-6>SBA-16>SBA-15, while those for the modified materials followed the order: SBA-15/DAPTES>KIT-6/DAPTES>SBA-16/DAPTES. Among the prepared initial samples, KIT-6 possessed the highest specific surface area. The nitrogen physisorption measurements of SBA-15 and SBA-16 showed a surface area of 880 m^2^/g and 890 m^2^/g and a pore volume of 1.00 cm^3^/g and 0.53 cm^3^/g, respectively. The significant decrease in the textural parameters, such as specific surface area, total pore volume, and pore diameter of the modified materials, indicate pore filling by DAPTES. This effect is more pronounced for SBA-16/DAPTES than for the SBA-15/DAPTES and KIT-6/DAPTES samples. Probably, this is due to the three-dimensional channel system and uniform-sized pores of the super large, cage-like structure of the SBA-16 with small pore sizes around 4.9 nm. Because of the structural peculiarities of the SBA-16, some pores of the modified material are blocked with the grafted bulk molecules. The specific surface area of the modified KIT-6/DAPTES decreased less than that of modified SBA-16/DAPTES. This might be due to the interpenetrating cylindrical pores, which favors the deposition of DAPTES in the pores of the SBA-16. The results from the N_2_ physisorption correspond very well with the TG and IR data, indicating grafting of DAPTES into the pores of the silica materials. The preservation of the mesoporous structure of all the used mesoporous silicas during the modification procedure was observed, which is in agreement with the XRD patterns in the low-angle range (Appendix A).

### 3.3. CO_2_ Adsorption Measurements

The mesoporous silicas (SBA-15, SBA-16, and KIT-6) modified with DAPTES were tested in dynamic conditions at atmospheric pressure. The results are presented in Table 2 and Figure 5. A higher amount of adsorbed CO_2_ was detected on the modified mesoporous silicas than on the initial ones. This effect is predetermined by the type of the adsorption sites, which, in the case of the initial silica materials, are the silanol groups. Quantitative ^29^Si NMR measurements were performed to evaluate the amount of silanol groups in the three silica samples. The quantitative single-pulse ^29^Si spectra (Appendix A) demonstrated the presence of a higher number of silanol groups in SBA-16 as compared to KIT-6 and SBA-15 (Appendix A). A correlation between the adsorption capacity and the number of SiOH groups was observed. Among the sorbents studied, SBA-16 displayed the highest value of the CO_2_ adsorption capacity, while the formation of a smaller number of SiOH groups in the SBA-15 and KIT-6 silicas led to lower adsorption of CO_2_ on them. In addition, the time needed for achieving the total adsorption capacity for the SBA-15 silica (T = 8 min) was shorter in comparison to the time determined for CO_2_ adsorption on to the SBA-16 and KIT-6 silica (T = 17 min) due to the different pore structures. Thus, the structure peculiarities of the three-dimensional pores of KIT-6, cage-like SBA-16, and two-dimensional pores in the hexagonal SBA-15 also influenced the performance of the sorbents in the CO_2_ adsorption experiments.

The modification with DAPTES led to an increase in CO_2_ adsorption capacity for the three silica materials (Table 2). The highest value for the CO_2_ adsorption capacity in dynamic conditions among the studied materials was detected for the modified SBA-15/DAPTES sample (3.9 mmol/g). The CO_2_ adsorption capacities of the samples decreased in the following order: SBA-15/DAPTES>SBA-16/DAPTES>KIT-6/DAPTES. The results showed that structural peculiarities of the mesoporous silicas significantly influenced the formation and localization of the adsorption sites. We assumed that the structural characteristics of SBA-15 led to the excellent CO_2_ uptake of 3.9 mmol/g due to its open structure that favors the modification with DAPTES (Figure 2). The total CO_2_ desorption was reached at 80 °C for the DAPTES-modified materials and at 40 °C for the parent ones. That implies strong interaction of the modified pore surface of the material, probably accompanied with chemisorption of CO_2,_ leading to the formation of bicarbonates that are electrostatically attracted to the positively charged ammonium groups (Figure 3). That was confirmed by the adsorption experiments performed with the addition of 1 vol.% water vapor to the CO_2_/N_2_ flow at a rate of 30 mL/min to determine the selectivity of the prepared materials for CO_2_ adsorption in the presence of water vapor, i.e., in a flow of CO_2_/H_2_O/N_2_. Interestingly, the modified samples showed higher adsorption capacities for CO_2_ in the presence of water vapor compared to that determined in a CO_2_/N_2_ flow, which is the opposite for the parent silicas. The higher selectivity for CO_2_ adsorption in the presence of water on the modified materials is most probably due to the chemisorption of CO_2_ on the adsorption sites (Table 2). Probably, the effect is enhanced by the presence of the highly polar P=O groups and the electron-rich aromatic structures.

Results on the CO_2_ capture on functionalized SBA-type adsorbents have been reported by other research groups. An adsorption capacity of 0.72 mmol/g under 101.32 kPa at 60 °C was determined for SBA-16 modified with N-(2-aminoethyl)-3-aminopropyltrimethoxysilane. The adsorbent displayed a high hydrothermal stability, and the combustion of the amino-groups in air was observed at temperatures above 200 °C [33,54]. Hiyoshi et al. [31] reported that the adsorption capacity of amino-silica (SBA-15) reached 1.8 mmol/g under 15 kPa CO_2_ and at 60 °C. It was found that the efficiency of adsorption increased on increasing the surface density of amine. Triamine-functionalized SBA-15 had a greater CO_2_ retention capacity value (3.12 mmol/g) than the one grafted with mono- and di-amine matrices. The presence of moisture enhanced the adsorbent performance, suggesting the participation of water molecules in the CO_2_ retention [55]. The modified materials studied in the present investigation show comparable or higher and more stable adsorption capacities than that for similar sorbents as described above.

The adsorption capacities of the synthesized nanomaterials were determined using a laboratory-scale fixed-bed reactor. The Yoon–Nelson model [56] has been applied for the investigation of adsorption kinetics in a fixed-bed column, and the linear form of the model is represented by Equation (1):ln(C_t_/C_o_ − C_t_) = κ_YN_t − τκ_YN_(1)
where κ_YN_ is the Yoon–Nelson rate constant (min^−1^), τ is the time required for 50% of adsorbate breakthrough (min), t is the sampling time (min), C_0_ is the initial concentration of CO_2_, and C is the concentration of CO_2_ at any time during evaluation. We found that the model and experimental data had a strong correlation (R^2^ > 0.99) (Appendix A).

The initial and the modified silicas were studied in five adsorption/desorption cycles to check their adsorption potential and selectivity for CO_2_ adsorption in the presence of water, which is very important for their practical application. The results show stable performance with preserved high adsorption capacity and selectivity for CO_2_ in five adsorption/desorption cycles for all the studied samples (Table 2).

The calculated heats of adsorption using the Clausius–Clapeyron equation for all samples based on the CO_2_ adsorption at 0 °C, 25 °C, and 50 °C are presented in Figure 6. The calculated heat of adsorption values for all samples were in the interval of 20–52 kJ/mol. Higher values were determined for the DAPTES-functionalized silicas in comparison to their parent analogs. The highest heat of adsorption (around 50 kJ/mol) was displayed by DAPTES-modified SBA-15 and SBA-16, which was two times higher than that of the initial SBA-15 and SBA-16 (25–27 kJ/mol). DAPTES-functionalized SBA-15 and SBA-16 silicas also exhibited a steep decrease in the heats of adsorption as a function of the adsorbed amount of CO_2_ as compared to the initial silicas, which indicates the presence of different adsorption sites for effective CO_2_ adsorption in them. Significantly lower heats of adsorption were detected for the KIT-6/DAPTES material (34 kJ/mol), which can be explained by the lower extent of DAPTES modification and corresponds to the low CO_2_ adsorption (Table 2).

The high CO_2_ isosteric heats of adsorption of the DAPTES-modified SBA-15 and SBA-16 materials led to increased CO_2_ uptake and selectivity (Table 2). The obtained results can be related to the appropriate structure of the large-pore mesoporous silicate used, the advantage of which is not only due to the more exposed DAPTES fragments but also due to the provided necessary path for CO_2_ diffusion.

## 4. Conclusions

The DAPTES-modified mesoporous silicas with different pore structures, 2D SBA-15, 3D SBA-16, and KIT-6, were successfully prepared using a simple two-step post-synthesis procedure. The mesoporous structure of the silica materials was preserved during the chemical grafting of the new DOPO derivative. The DAPTES-modified materials showed a higher capacity for CO_2_ capture in comparison to the parent ones. The adsorption capacity values depended on the structural peculiarities of the supports used. CO_2_ capture is favored by the presence of polar groups and electron-rich fragments in the structure of the grafted agent. Among the studied sorbents, the DAPTES-modified mesoporous SBA-15 silica showed the highest adsorption capacity for CO_2_ (3.9 mmol/g). Enhanced CO_2_ adsorption capacities were detected for the functionalized silicas in the experiments with 3 vol.% CO_2_ plus 1 vol.% water vapor due to the facilitated formation of chemisorbed CO_2_ in the form of a bicarbonate ion. The chemisorption of CO_2_ for DAPTES-modified silicas was assumed based on the calculated heats of adsorption for the studied materials. Total CO_2_ desorption from the modified materials was achieved at 80 °C. The functionalized silica supports displayed good performance parameters such as high adsorption capacity, stability, and low energy consumption for CO_2_ desorption, which is worth being considered for further studies and scaling experiments. The experimental data can be appropriately described by the Yoon–Nelson kinetic model.

## Data Availability

Data are contained within this article and the Appendix A.

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
