# Peer review of "Engineering of Silica Mesoporous Materials for CO2 Adsorption"

_materials, 2023, doi:10.3390/ma16114179_

Round 1

Reviewer 1 Report

The manuscript submitted by Koseva et al. contains a study about different kinds of mesoporous silica (i.e., SBA-15, SBA-15, KIT-6) that are functionalized utilizing DAPTES. The hydrogen phosphonate moiety that is therefore introduced on the surface is improving the CO2 adsorption capacity of the porous silica. The largest CO2 uptake is observed for the SBA-15 silica derivative.

Despite the topic being interesting for global challenges as CO2 adsorption I am afraid the study lacks some serious details as explained below. Please compare your materials with state of the art/comparable materials from the literature. More detailed comments are listed below.

-          I recommend not using abbreviations in the abstract.

-          ll. 31: “Due to presence of greenhouse gases in the atmosphere the average temperature of the Earth is positive and friendly for the living systems.”

I am afraid this sentence is confusing to the reader. The manuscript would benefit from a native speaker for proofreading.

-          ll. 76-78: “9,10-Dihydro-9-oxa-10-phosphaphenanthrene-10-oxide (DOPO) has been selected as modifying molecule due to its reactivity and ability to improve the anti-oxidative and flame-retarding properties of materials [36].” How does the aspect of DOPO relate to CO2 adsorption an motivates the authors to this study?

-          Ll95-117: Please include heating ramp and atmosphere for the calcination procedure.

-          Ll 163-165: “[...]by nitrogen adsorption at -196 °C in the pressure range p/p0 = 0.001–1, using an analyzer “AUTOSORB iQ-MP/AG” (Anton Paar GmbH, Austria), with a heating rate of 5 °C/min in airflow”

N2 sorption experiments typically reach close to a relative pressure of 1 but do not reach 1. Do the heating rate and airflow belong to the degassing/activation of the samples?

-          Figure 2: Why do sample SBA-15/DAPTES and SBA-16/DAPTES exhibit “negatives bands” (bands are facing upwards towards larger values of Transmittance)? Please make sure that the baseline correction was performed correctly. The horizontal axis should be wavenumber not wavelength. The IR spectra do not prove the functionalization of the silica surface. A physical mixture of functionalization agent and silica would lead to the same spectra. Furthermore, how did you secure that DAPTES did only react on the silica surface? It contains three ethoxy groups and is able to react with itself. Please provide proof that the functionalization was successful.

-          How stable are the silica samples temperature wise? The authors are talking about thermogravimetric data but those are not shown. Please include all of them for pure silica and functionalized silica in the supplementary information section.

-          l.263 Consider citing the IUPAC paper you are referring to here.

-          Figure 3: SBA-15/DAPTES shows only the adsorption branch. Please include a full isotherm and furthermore include the diffraction patterns of the silica phases. The maximum of the size distribution is mentioned on Table 1, but the authors do not show the pore size distribution. Please include it and mention how it was determined (BJH, DFT, other)? Additionally, explain in which relative pressure range the BET surface area and the total pore volume was determined.

-          Why does the pore size of KIT decrease 0.9 nm after functionalization, but the pore size of the other silica samples does not change or only about 0.4 nm?

-          ll.297-298: “The formation of a smaller amount of SiOH groups in the SBA-15 and KIT-6 silicas leads to lower adsorption of CO2  on them.” Please quantify the amount of SiOH groups for each silica sample to prove your claim.

-          ll.301-305 “This result could be explained by the structure peculiarities i.e. the interpenetrating network of the three-dimensional pores of KIT-6 and cage–like SBA-16 hinder the access to the adsorption sites, which is the opposite for the more open two-dimensional pores in the hexagonal SBA-15.” In the literature, KIT-6 silica is typically referred to have more accessible pores (compared to SBA-15 and -16). Please discuss your claims in detail.

-          Scheme 3: In the Scheme the authors describe how they think that the functionalization interacts with CO2. Is there any proof for this or is it speculation?

-          Figure 5 and discussion: the measurement of heat of adsorption is performed for two different temperatures. Please add a third temperature so that you are able to include error values for the heat of adsorption fit. The heat of adsorption values of the silica are very different, please explain why. The calculated heat of adsorption over volume should be plotted as a point diagram (and not a line because it is not a continuous measurement). Also, the “volume” axis is labeled with mmol/g.

Reviewer 2 Report

Review of “Engineering of silica mesoporous materials for CO2 adsorption” by Oyundari Tumurbaatar, Margarita Popova, Violeta Mitova, Pavletta Shestakova and Neli Koseva

This paper presents adsorbents materials based on silica for CO2 adsorption. Preparation of the adsorbents as well as their functionalization’s were made. Then, characterization of the materials was accomplished by means of NMR, FTIR, TGA, N2 adsorption/desorption isotherms. Then CO2 adsorption capacity was measured in different conditions (other compounds and concentrations). The paper must be revised prior to its publication in Materials journal concerning the following issues:

1.      Authors proved a stream containing water, and they prove whether CO2 capacity is inhibited or not, but what about water solubility of the materials? Have you proved different adsorption cycles in those conditions in order to ensure that CO2 adsorption capacity is preserved?

2.      TGA measurements were made but I could not find the TGA figures both in the manuscript or Supplementary Material.

3.      Authors must give CO2/N2 and CO2/H2O selectivity in order to compare with previous results of the literature.

4.      It is well known that CO2 adsorption capacity is improved in microporous materials and you stated and proved that your silica materials are mainly mesoporous one. Could you explain that?

5.      It is important to relate CO2 adsorption capacity with some properties such as porosity, or microporosity.

6.      A table comparing the CO2 adsorption capacities obtained and selectivities with other previously reported in the literature is essential.

7.      Breakthrough curves are not well defined. May you short your method in order to have points at least each 1 min? In this sense, what about kinetics of the process? You can fit to well-known models such as Yoon & Nelson and try to explain the differences found between the prepared materials and the literature.

8.      What about regeneration and recyclability? I think authors should present cycles for the materials to demonstrate their reusability.

Reviewer 3 Report

The manuscript develops a series of DAPTES functionalized mesoporous SiO2 for CO2 capture. Before the paper is published, there are some mistakes need to be modified.

1. There are some grammatical mistakes in the paper that need to be corrected. For example line 12 attract needs to be changed to attracts.

 2. What is the content of DAPTES in the mesoporous SiO2?

 3. How do you make sure that DAPTES is successfully grafted onto the mesoporous SiO2 rather than adsorbed on it?

Reviewer 4 Report

The "Engineering of Silica mesoporous materials for CO2 adsorption" manuscript presents a comparison of properties between three "raw" adsorbents and the same three ones modified.

Although this is an interesting and extensively working in the materials modification, the applicability of the materials and the performed modifications do not present high level of novelty in the CO2 adsorbents production area.

In my opinion, some revisions should be done before acceptance. 

- Introduction: need to be improved; some published works using SBAs materials and SBAs with modifications for CO2 capture should be reported; 

- Materials and Methods: experiences with the materials modified were done with materials in powder form? How to due the dynamic experiments? I think that the point 2.5.2 should be more detailed;

- Results and discussion:

Figure 3 must present the results using different symbols for each material; in black-white mode, it is not easy to identify the correspondent material.

Point 3.3 CO2 measurements, published works should be used to compare with these results.

Reviewer 5 Report

The paper titled “Engineering of silica mesoporous materials for CO2 adsorption” reports the results on the preparation of a new derivative of (3-aminopropyl)triethoxysilane and its use in the post-synthetic modification of a number of mesoporous silicas. In order to improve the anti-oxidative and flame-retarding properties of materials, 9,10-Dihydro-9-oxa-10-phosphaphenanthrene-10-oxide has been used as modifying molecule. I recommend this paper for publication after minor revision. The comments are listed below.

1.      All the abbreviations, including DOPO, should be defined when first used.

2.      Punctuation needs to be revised.

3.      The Introduction can be improved by considering the following paper: Iugai et al. MgO/Carbon Nanofibers Composite Coatings on Porous Ceramic Surface for CO2 Capture, Surface and Coatings Technology. 2020. V.400. 126208:1-7. DOI: 10.1016/j.surfcoat.2020.126208.

4.      All over the text: “hour”, “hours” -> “h”

5.      All over the text: “l”, “ml” -> “L”, “mL”

6.      Figure 1: spectrum (a) should come before spectrum (b).

7.      Table 1: upper and lower indexes should be checked.

8.      All over the text: “in dynamic conditions” -> “under dynamic conditions”

9.      Figure 5: The graphs should be labeled as (a), (b) …; the corresponding description should be given in the caption. Note that not only the heats are presented.

Round 2

Reviewer 1 Report

The authors revised the manuscript and added NMR measurements indicating that the silica samples contain different amounts of silanol groups and that the grafting process with DAPTES was successful. Furthermore, they added more CO2 adsorption data, but the discussion of the heat of adsorption is confusing and probably misleading. Therefore, I recommend a major revision again. Please find more detailed comments below.

Remark 9: How stable are the silica samples temperature wise? The authors are talking about thermogravimetric data but those are not shown. Please include all of them for pure silica and functionalized silica in the supplementary information section.

Answer: The TG curves (Figure S9) for all studied samples (initial and modified silica materials) have been included in the Supplementary Material. A brief discussion has been included in p. 7.

Reply reviewer: The samples were degassed at 80 °C prior N2 physisorption analysis and 150 °C prior CO2 adsorption measurements. The thermogravimetric analysis in Figure S8 exhibits temperatures between about 150 °C – 600 °C (afterward, isothermal conditions). I am interested in a measurement that starts at room temperature to show that the functionalization is stable at 80 °C and 150 °C.

Remark 11: Figure 3: SBA-15/DAPTES shows only the adsorption branch. Please include a full isotherm and furthermore include the diffraction patterns of the silica phases. The maximum of the size distribution is mentioned on Table 1, but the authors do not show the pore size distribution. Please include it and mention how it was determined (BJH, DFT, other)? Additionally, explain in which relative pressure range the BET surface area and the total pore volume was determined.

Answer: The desorption branch of the isotherms of SBA-15/DAPTES has been added. We apologize for the mistake and the needed correction has been made. The figure about pore size distribution has been also added in Figure 4 (new numbering). Additional information about the used method for calculation of surface area and pore size has been provided.

Reply Reviewer: The changes in the revised manuscript regarding the analysis of the isotherms and pore size distribution are appreciated. However, the footnote in Table 1 contains misleading information. The footnote to the surface area (SBET) states that the surface area is determined by “BJH method”. This is a mix-up; the footnote must be placed in the 4th column (pore diameter).

The pore size distributions (BJH) are shown in Figure 4b. When using the BJH approach, the ordinate typically shows the change of pore volume with respect to the change in pore size (dV/dD or sometimes dV/dlog(D)). The authors plot the pore volume versus the pore diameter. Also, the pore volume values are pretty large. Please elaborate on the plot and revise it if needed. I recommend using a line-symbol plot to indicate that the measurement is not continuous.

The authors do not provide the diffraction X-ray diffraction patterns despite being requested. Therefore, the authors do not have any proof that they successfully synthesized the silica phases.

Remark 12: Why does the pore size of KIT decrease 0.9 nm after functionalization, but the pore size of the other silica samples does not change or only about 0.4 nm?

Answer: Thank you for drawing the attention to that difference. We recalculated the pore size of KIT-6/DAPTES taking into account the broadened peak describing the pore size distribution. The half width of the peak was used for a more precise calculation of pore diameter in this case and the new value is presented in Table 1.

Reply reviewer: I am surprised that the pore size of the functionalized KIT-6 sample was analyzed differently compared to the other samples. Furthermore, I do not find the explanation in the manuscript about how the half-width of the peak was utilized for a “more precise calculation of the pore diameter”. Therefore, please carefully revise the analysis and the interpretation of the data accordingly.

Remark 16: Figure 5 and discussion: the measurement of heat of adsorption is performed for two different temperatures. Please add a third temperature so that you are able to include error values for the heat of adsorption fit. The heat of adsorption values of the silica are very different, please explain why. The calculated heat of adsorption over volume should be plotted as a point diagram (and not a line because it is not a continuous measurement). Also, the “volume” axis is labeled with mmol/g.

Answer: Additional measurements were done and the calculated heats of adsorption in Figure 5 are based on the measurements at 3 temperatures. The needed corrections have been made.

Reply reviewer: The ordinate is still labeled as “adsorbed volume” with the units “mmol/g”. mmol is not a volumetric unit.

The heat of adsorption measurements/calculations shown in Figure 6 are confusing. In the revised version manuscript, changes are marked in yellow, and the curve for KIT-6 at T=50 °C is also yellow, so it cannot be seen (except at low pressure parts where it overlaps with the other measurements). When calculating the isosteric heat of adsorption, a minimum of two isotherms at different temperatures are needed to calculate the heat of adsorption. Because a linear fit (i.e., Clausius-Clapeyron equation) through two data points is always a straight line that hits both points perfectly, it is impossible to evaluate a potential error value. Therefore, I suggested using a third isotherm (at a third temperature) to evaluate the heat of adsorption and including error values. Clausius-Clapeyron equation assumes that the heat of adsorption is constant over the investigated temperature range, which is approximately true if a small temperature window is used. The authors used temperatures of 0 °C, 25 °C and 50 °C and then calculated heat of adsorption for 25 °C and 50 °C. The largest values were reported for SBA-15/DAPTES at 25 °C (around 80 kJ/mol; around 27 kJ/mol for pristine SBA-15)). KIT-6/DAPTES at 25 °C exhibits a heat of adsorption of around 22 kJ/mol, and the pristine KIT-6 has similar values at least around an adsorbed amount of 0.1 mmol/g of CO2. The authors state in the manuscript that “Among them the highest heats of adsorption (80-85 kJ/mol) displayed modified SBA-15 favored by the more open cylindrical pores of the sorbent and  consequently stronger interactions between DAPTES residues and CO2 molecules. Similar heats of adsorption were detected for KIT-6 and KIT-6/DAPTES, most probably due to the hindered access of CO2 molecules in the partially blocked pores of KIT-6 upon DAPTES grafting.” How are the values similar? From my point of view, the absolute values are vastly different, and the trend of the heat of adsorption is, too. The pristine KIT-6 sample has a larger heat of adsorption compared to the functionalized one (at 25 °C) at adsorbed amounts of CO2 larger than approximately 0.1 mmol/g. Also, the explanation of pore accessibility of the different silica species seems not to agree with the arguments discussed for the initial N2 physisorption data. Why is the heat of adsorption larger for KIT-6/DAPTES at 50 °C compared to 25 °C, and why is the trend not the same for all silica types? A more detailed discussion and probably more measurements are needed to prove the claims stated in the manuscript.

Reviewer 2 Report

The paper can be published.

Author Response

The authors thank the reviewer for the comments.